# Power Laws and Elementary Particle Decays

**Leonardo Chiatti** 

ASL VT Medical Physics Laboratory; Via Enrico Fermi 15, 01100 Viterbo, Italy; leonardo.chiatti@asl.vt.it

**Abstract:** This study analyzes the correlation between the lifetime and the rest energy of the unstable particle states with a lifetime greater than the zeptosecond ($10^{-21}$ s), using data available from the Particle Data Group. This set of states seems to be divided into three groups, in each of which the two quantities can be correlated through a remarkably accurate power law. Although this fact does not represent anything new compared to the predictions of the Standard Model, it nevertheless reveals an unexpected order structure in the set of particle decays, emerging from such predictions.

**Keywords:** elementary particle decays; emergence; power laws

---

## 1. Introduction

Consider an unstable particle of rest mass $M$ and lifetime $\tau$ (half-life $T = \tau\ln 2$); we perform the transformation of these two quantities into two new quantities (the time scale $\theta$ and the dimensionless parameter $n$) through the relations:

$$\frac{\hbar}{\theta}2^{-n} = \frac{\hbar}{T} \tag{1}$$

$$\frac{\hbar}{\theta n} = Mc^2 \tag{2}$$

The relevance of these transformations is due to the fact that it is possible to subdivide the unstable elementary particles with lifetime greater than the zeptosecond (1 zs = $10^{-21}$ s) into three groups in each of which the following power laws hold, with slightly different values of the parameters:

$$\frac{\theta}{\theta_0} = A\left(\frac{Mc^2}{E_0}\right)^{-B} \tag{3}$$

$$n = \frac{1}{A\left(\frac{Mc^2}{E_0}\right)^{1-B}} \tag{4}$$

Equations (3) and (4) are deductible from each other taking into account Equations (1) and (2). Parameters $A$ and $B$ are positive. The time scale $\theta_0$ is completely arbitrary and we choose it equal to the time taken by the light to travel the classic radius of the electron, so as to express $A$ with a comfortable value; $E_0 = \hbar/\theta_0$. It is evident that starting from $M$ it is possible, through (3) and (4), to derive both $\theta$ and $n$, and then $T$. It is therefore possible to derive $T$ from $M$ through auxiliary parameters, while it does not appear possible, as we will show, to derive directly $T$ as a function of $M$. Relations (3) and (4) therefore express a relation between lifetimes and masses.

The relevance of (3) and (4), which constitute the main result of this work, lies in the fact that their existence seems to be completely unexpected. The mean life of each individual decay process (and consequently the total lifetime of a particle, which is the harmonic average of the mean lives of the single decay processes to which it is subject) is completely calculable starting from the Standard Model

or its approximations. The calculation requires, as input data, the masses of the final states, which constrain the phase space accessible to each process. Therefore it should in principle be impossible to find an empirical relationship between the lifetime of a particle and its mass only. On the other hand, this is exactly what Relations (3) and (4) do. The explanation of the paradox is not hidden in some "new physics" because all the processes of decay, whether weak, electromagnetic or strong, can be calculated from the Standard Model and therefore, (3) and (4) do not add nothing of new to this knowledge. What (3) and (4) say is instead that as the mass increases, phase space factors and decay modes tend to combine in a way that leads to the formation of power laws.

A remarkable fact is that although hadrons and leptons are very different classes of particles, (3) and (4) apply equally successfully to both. Here too, the paradox disappears if one bears in mind that, according to the Standard Model, the components of the hadrons subject to decay (the quarks) interact with the weak field in a manner very similar to the leptons. And the leptons can appear as final states in the weak decays of hadrons. The involvement of hadrons and leptons in a common emergent order structure is therefore interpretable as an effect of the similarity of their role in the generation of this structure.

From (1) and (2), the following relation is obtained:

$$\frac{\hbar}{Mc^2 T} = \frac{n}{2^n} \tag{5}$$

This relation was proposed, as a definition of the parameter $n$, by previous researchers some decades ago [1–5]. These authors also emphasized aspects of duality between $M$ and $T$ in the decay of elementary particles. However, we propose here an entirely different argument based on empirical Relations (3) and (4). Equation (5) is however suggestive for the following reason. Let us imaging preparing, at instant 0, an unstable particle of mass $M$ and half-life $T$; the probability that it is still in the initial state at time $t = nT$ is given by $p(t) = 2^{-n}$. Let us now consider the particular instant $t$ to which the following relation holds:

$$p(t) = \frac{\hbar}{Mc^2} \cdot \frac{1}{t} \tag{6}$$

That is, the probability of the unstable state equals the ratio of the Compton time of the state to the total time elapsed since the preparation of the particle in that state. The (6) is clearly a definition of $n$ and it coincides with (5). The information $I$ obtained with the confirmation that the particle is still in the unstable state at this particular instant is then:

$$I = -\log_2 p(t) = \log_2\left(\frac{\hbar}{Mc^2 T n}\right) = n \tag{7}$$

As is immediately verified by inserting (5) in the last step of (7). $n$ is therefore a particular measure of information on the state. The right-hand side of (5) is the corresponding Shannon average entropy. These considerations may perhaps be useful for linking Transformations (1) and (2) to a physical mechanism. But in this brief note we will not go into this aspect in greater depth.

The work is articulated as follows. Section 2 briefly exposes the systematics of elementary particle decays; in particular, the absence of a monodromic function that correlates half-life and mass is highlighted. In Section 3, Relations (3) and (4) and the fitting of experimental data are discussed. Some general arguments are discussed in the final section.

All the particle data used in this study are from Particle Data Group (PDG) 2018 [6].

## 2. The Decays of Elementary Particles

The unstable particles with a lifetime greater than the zeptosecond are listed in Table 1; the decays of these particles occur essentially through weak or electromagnetic channels. The plot in Figure 1 shows the representative points of these particles. On the abscissa, the lifetime is reported in seconds,

while on the ordinate, no quantity is reported; the vertical axis is introduced only to facilitate the visualization of the spacing of the points.

**Table 1.** Unstable particles with lifetime greater than the zeptosecond.

|   | Particle | Mass (MeV) | Measured Width (MeV) | $n$ (Equation (5)) | $n$ (Equation (4)) | Calculated Width (MeV) | Calc/Meas Width |
|---|---|---|---|---|---|---|---|
| 1 | $n$ | $9.40 \times 10^2$ | $7.43 \times 10^{-25}$ | 96.11 | | | |
| 2 | $\mu$ | $1.06 \times 10^2$ | $3.00 \times 10^{-16}$ | 63.78 | 64.76 | $3.17 \times 10^{-16}$ | 1.06 |
| 3 | $\pi^\pm$ | $1.40 \times 10^2$ | $2.53 \times 10^{-14}$ | 57.64 | 63.51 | $9.77 \times 10^{-16}$ | $3.86 \times 10^{-2}$ |
| 4 | $K_{0L}$ | $4.98 \times 10^2$ | $1.27 \times 10^{-14}$ | 60.53 | 58.11 | $1.34 \times 10^{-13}$ | $1.06 \times 10$ |
| 5 | $K^\pm$ | $4.94 \times 10^2$ | $5.33 \times 10^{-14}$ | 58.41 | 58.14 | $1.30 \times 10^{-13}$ | 2.44 |
| 6 | $\Xi_0$ | $1.31 \times 10^3$ | $2.27 \times 10^{-12}$ | 54.3 | 54.31 | $4.61 \times 10^{-12}$ | 2.03 |
| 7 | $\Lambda$ | $1.12 \times 10^3$ | $2.50 \times 10^{-12}$ | 53.92 | 54.91 | $2.63 \times 10^{-12}$ | 1.05 |
| 8 | $\Xi^-$ | $1.32 \times 10^3$ | $4.02 \times 10^{-12}$ | 53.47 | 54.28 | $4.73 \times 10^{-12}$ | 1.18 |
| 9 | $\Sigma^-$ | $1.20 \times 10^3$ | $4.46 \times 10^{-12}$ | 53.17 | 54.64 | $3.37 \times 10^{-12}$ | $7.55 \times 10^{-1}$ |
| 10 | $\Omega$ | $1.67 \times 10^3$ | $8.02 \times 10^{-12}$ | 52.79 | 53.39 | $1.09 \times 10^{-11}$ | 1.36 |
| 11 | $\Sigma^+$ | $1.19 \times 10^3$ | $8.25 \times 10^{-12}$ | 52.24 | 54.67 | $3.27 \times 10^{-12}$ | $3.96 \times 10^{-1}$ |
| 12 | $K_{0S}$ | $4.98 \times 10^2$ | $7.38 \times 10^{-12}$ | 51.12 | 58.11 | $1.34 \times 10^{-13}$ | $1.82 \times 10^{-2}$ |
| 13 | $B^\pm$ | $5.28 \times 10^3$ | $4.28 \times 10^{-10}$ | 48.59 | 49.26 | $5.57 \times 10^{-10}$ | 1.30 |
| 14 | $B_0$ | $5.28 \times 10^3$ | $4.39 \times 10^{-10}$ | 48.56 | 49.26 | $5.57 \times 10^{-10}$ | 1.27 |
| 15 | $B_{0S}$ | $5.38 \times 10^3$ | $4.92 \times 10^{-10}$ | 48.41 | 49.19 | $5.93 \times 10^{-10}$ | 1.21 |
| 16 | $\Lambda_{0b}$ | $5.64 \times 10^3$ | $6.16 \times 10^{-10}$ | 48.15 | 49.03 | $6.93 \times 10^{-10}$ | 1.13 |
| 17 | $B_{c+}$ | $6.29 \times 10^3$ | $1.42 \times 10^{-9}$ | 47.07 | 45.59 | $7.82 \times 10^{-9}$ | 5.51 |
| 18 | $D^\pm$ | $1.87 \times 10^3$ | $6.24 \times 10^{-10}$ | 46.49 | 44.50 | $4.84 \times 10^{-9}$ | 7.75 |
| 19 | $D_s{}^\pm$ | $1.97 \times 10^3$ | $1.41 \times 10^{-9}$ | 45.36 | 44.54 | $4.94 \times 10^{-9}$ | 3.50 |
| 20 | $\Xi_c{}^+$ | $2.47 \times 10^3$ | $1.88 \times 10^{-9}$ | 45.26 | 44.74 | $5.41 \times 10^{-9}$ | 2.88 |
| 21 | $D_0$ | $1.86 \times 10^3$ | $1.59 \times 10^{-9}$ | 45.09 | 44.49 | $4.83 \times 10^{-9}$ | 3.04 |
| 22 | $\tau$ | $1.78 \times 10^3$ | $2.23 \times 10^{-9}$ | 44.52 | 44.45 | $4.74 \times 10^{-9}$ | 2.13 |
| 23 | $\Lambda_c{}^+$ | $2.29 \times 10^3$ | $3.30 \times 10^{-9}$ | 44.31 | 44.68 | $5.25 \times 10^{-9}$ | 1.59 |
| 24 | $\Xi_{c0}$ | $2.47 \times 10^3$ | $5.85 \times 10^{-9}$ | 43.57 | 44.74 | $5.41 \times 10^{-9}$ | $9.25 \times 10^{-1}$ |
| 25 | $\Omega_{c0}$ | $2.70 \times 10^3$ | $9.50 \times 10^{-9}$ | 43 | 44.82 | $5.61 \times 10^{-9}$ | $5.90 \times 10^{-1}$ |
| 26 | $\pi_0$ | $1.35 \times 10^2$ | $7.85 \times 10^{-6}$ | 28.38 | 24.65 | $1.82 \times 10^{-4}$ | $2.32 \times 10$ |
| 27 | $\eta$ | $5.47 \times 10^2$ | $1.20 \times 10^{-3}$ | 22.84 | 22.67 | $2.69 \times 10^{-3}$ | 2.24 |
| 28 | $\Sigma_0$ | $1.19 \times 10^3$ | $8.91 \times 10^{-3}$ | 20.95 | 21.63 | $1.14 \times 10^{-2}$ | 1.28 |
| 29 | $\gamma(3S)$ | $1.04 \times 10^4$ | $2.63 \times 10^{-2}$ | 22.62 | 18.99 | $5.46 \times 10^{-1}$ | $2.08 \times 10$ |
| 30 | $\gamma(2S)$ | $1.00 \times 10^4$ | $4.40 \times 10^{-2}$ | 21.77 | 19.04 | $5.10 \times 10^{-1}$ | $1.16 \times 10$ |
| 31 | $\gamma(1S)$ | $9.46 \times 10^3$ | $5.25 \times 10^{-2}$ | 21.41 | 19.10 | $4.63 \times 10^{-1}$ | 8.82 |
| 32 | $J/\psi(1S)$ | $3.10 \times 10^3$ | $8.80 \times 10^{-2}$ | 18.88 | 20.42 | $6.49 \times 10^{-2}$ | $7.38 \times 10^{-1}$ |
| 33 | $D^{*\pm}$ | $2.01 \times 10^3$ | $9.60 \times 10^{-2}$ | 18.08 | 20.96 | $2.97 \times 10^{-2}$ | $3.10 \times 10^{-1}$ |
| 34 | $J/\psi(2S)$ | $3.69 \times 10^3$ | $3.37 \times 10^{-1}$ | 17.06 | 20.21 | $8.86 \times 10^{-2}$ | $2.63 \times 10^{-1}$ |

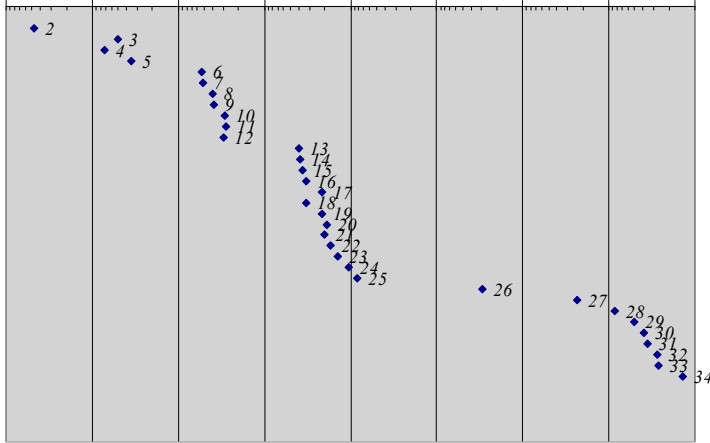

**Figure 1.** Particles with lifetimes greater than the zeptosecond.

The numbering of each point is explained in Table 1. Point 1, i.e., the neutron, is not reported due to its exceptionally long lifetime: the free neutron decays with emission of a $\beta^-$ with a lifetime of about 15 minutes. It is well known that the slowness of this decay is due to the small phase space factor: the mass of the neutron is almost equal to that of the final state of transformation, which is the proton. For this anomaly we will exclude the neutron from our subsequent considerations. The remaining decays are, as one can immediately see from the plot, grouped in a way dependent on the quark composition or on the lepton flavor. Point 2 is the muon; points 3–5 represent pseudoscalar mesons (including the $K_{0L}$) composed of quarks $u$, $d$, $s$; points 6–12 represent the baryons (more the $K_{0S}$) composed of quarks $u$, $d$, $s$; points 13–17 represent hadrons with the bottom ($b$) as the heavier quark; points 18–25 represent hadrons with charm ($c$) as the heavier quark; they also include tau. All these states decay through weak interactions, which induce the transmutation of the heavier quark into a quark of lower mass.

The decays of the particles 26–34 occur instead through electromagnetic (and strong) channels. Here too we can notice groupings. Point 26 is the neutral pion; points 27–28 are hadrons with the strange ($s$) as the heavier quark; points 29–31 are hadrons with the bottom ($b$) as the heavier quark; points 32–34 are hadrons with the charm ($c$) as the heavier quark.

To the right of the area represented in Figure 1, there is the domain of the particles that predominantly decay by strong interaction. The lifetimes of these particles are all less (or about equal) to the zeptosecond and no plausible grouping is seen: their spectrum is practically continuous. We do not report these particle states here, which are a multitude (over one hundred and twenty), because the existence of Relations (3) and (4) seems to be connected to well defined groupings.

Let us now focus our attention on particle states 2–34 and ask ourselves if it is possible to obtain the mass of a state simply knowing its lifetime. The correlation plot of masses and lifetimes is shown in Figure 2.

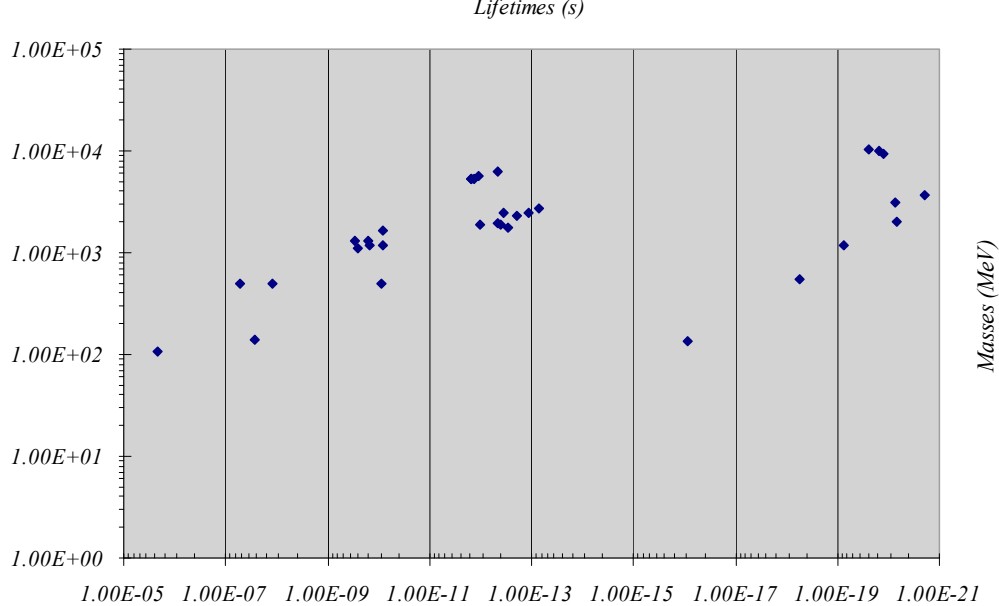

**Figure 2.** Correlation plot of the masses and lifetimes of states with lifetime greater than 1 zs.

The points show a certain clustering, easily correlated with the groups we have previously described. However, it is evident that there is not a sufficiently close correlation to allow one of the two parameters to be derived from the other through a monodromic function.

## 3. Power Laws

For each of states 2–34, it is possible to calculate $n$ through (5); knowing $n$, it is possible to calculate $\theta$ from (2). Having done this, it is possible to look for a correlation between $n$ (or $\theta$) and $M$. The unexpected result is that the states are divided into three groups within each of which this correlation, unlike that represented in Figure 2, is very narrow and admits a regression with power law of the types (3) and (4). The first group includes states 2–16; the second group, the states 17–25; and the third group, states 26–34. The best fit values of parameters $A$ and $B$ for each group and the respective correlation indices are as follows:

| | | | |
|---|---|---|---|
| states 2-16; | $A = 0.015$ | $B = 0.93$ | $r^2 = 0.998$ |
| states 17-25; | $A = 0.024$ | $B = 1.02$ | $r^2 = 0.995$ |
| states 26-34; | $A = 0.039$ | $B = 0.94$ | $r^2 = 0.991$ |

It is then possible to invert the reasoning, starting from (3) and (4) to calculate $n$ (or $\theta$) from $M$, and use (1) or (5) to determine $T$. The width of the state calculated in this way is shown in Table 1 and is compared with the one measured. As can be seen, the deviations are generally contained within an order of magnitude ($\pm$) and only in two cases (charged pion, $K_{0S}$) they reach the two orders. In any case, considering that the lifetimes of states 2–34 span about 15 orders of magnitude, it can be argued that the agreement is satisfactory.

In Figure 3 the quantity ($\theta/\theta_0$), obtained from the experimental data through (5) and (2), is plotted as a function of ($Mc^2/E_0$) in a doubly logarithmic scale. The representative points of the states 2–16, 17–25 and 26–34 are reported in three different colors and shapes (respectively diamonds, squares and triangles), in order to facilitate the distinction between families. The three fitting lines of the single families, Equation (3), are also reported. The reader can compare with Figure 2 and note the difference in correlation with the new variable.

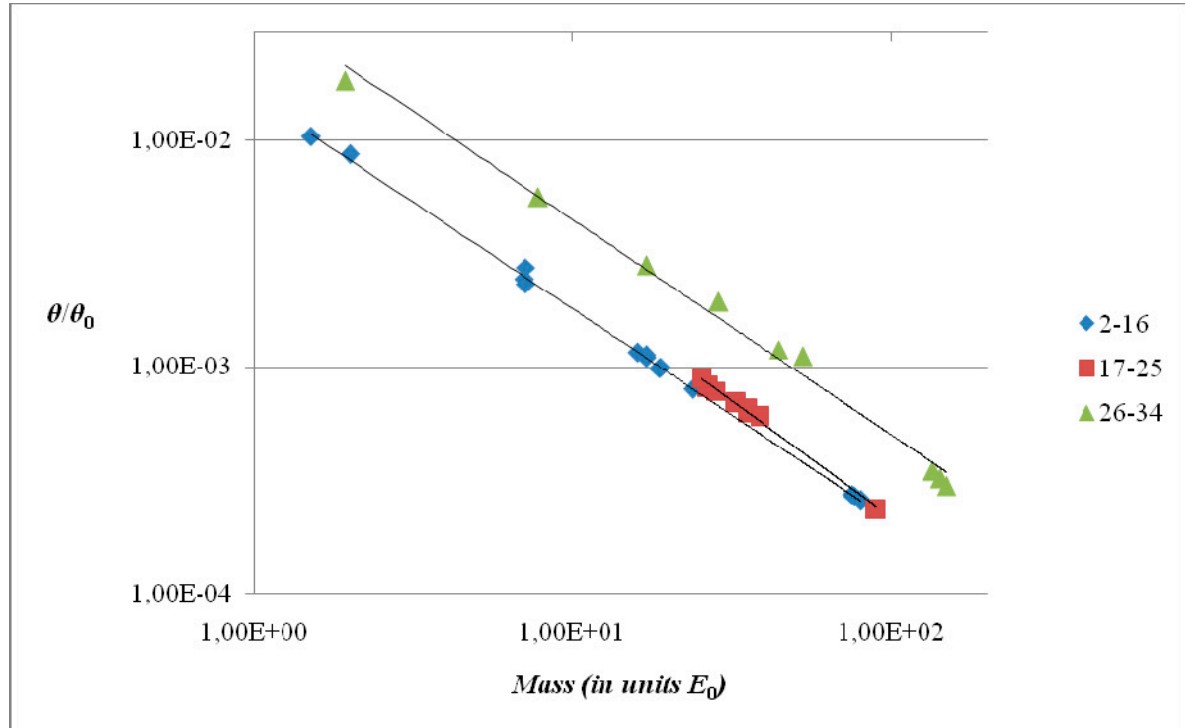

**Figure 3.** Correlation plot of $\theta$ and mass.

Physically, the first group includes all states without charm quark with a width less than that of the neutral pion. The second group includes all states with charm quark with a width less than that of the neutral pion. The third group includes the neutral pion and all states with a width greater than that of the neutral pion. In addition, the muon is included in the first group while the tau belongs to the second group. These two particles are leptons, and the applicability of the same power law to hadrons and leptons does not seem trivial. In this regard, we must keep in mind what we already said in the introduction about the emergence of (3) and (4) from the Standard Model. We return to this in the next section.

## 4. Discussion

Relations (3) and (4) constitute the main result of this note. Although, formally, they allow to express the half-life of the unstable particles (with lifetime greater than the zeptosecond) as a function of the mass, their interest does not lie in a rather improbable practical application. In fact, half-life can be estimated starting from the Standard Model or its approximations and in this respect, (3) and (4) do not add anything. Rather, it is their mere existence that raises questions. From a general point of view the calculation of the half-life requires the specification of the decay channels, and the result depends on the masses of the products, so that it seems impossible to express the half-life of a state only as a function of the mass of that same state. But this is what (3) and (4) do. This implies that taking into consideration increasingly massive states, the network of possible decays to lower mass states produces a total half-life that can be expressed as a function of the mass of the considered state through power laws. To our best knowledge, a mechanism of this type has never been described in the vast literature on the applications of the Standard Model. A further element of novelty is that with respect to the formation of the power laws mentioned, the particles are arranged in three different "families" with a combination mechanism that remains to be clarified. The third unexpected element is that this mechanism, whatever it is, does not distinguish between hadrons and leptons.

It is good to make it clear that what emerges from (3) and (4) does not in any way constitute "new physics" beyond the Standard Model. The situation seems rather to be similar to that in physical chemistry for what concerns the relationship between the primary structure of a protein and the tertiary or quaternary structures of the same folded protein. Even the elucidation of this relationship remains an unresolved problem. In the comparison, the primary structure is given by the local interactions between the amino acids of the sequence and it determines the higher order structures (under certain assigned conditions). Similarly, the description offered by the Standard Model is focused on the interactions between elementary fermions and gauge fields that determine the single decay process. But this description does not elucidate how the processes of decay combine with each other to form higher order patterns. However, as the understanding of protein folding does not require any "new chemistry", so the understanding of Equations (3) and (4) does not require "new physics". Another situation that shows a certain parallelism with the present one is described by the relationship between the Schröedinger equation for a polyelectronic atom and the periodic table, in which the chemical elements are ordered according to the atomic number. The order structure of the table cannot be derived by solving the wave equation relative to an atom with a specified atomic number, but only by comparing the solutions of the equations relating to different atomic numbers. In the present case the wave equations are the calculations of $T$ based on the Standard Model for each single unstable particle, while the role of the "atomic number" is played here by the parameters $n$, $\theta$. The difference is that while the atomic number is immediately reflected in the structure of the wave equations, the parameters mentioned do not seem to have evident correlates in the calculations of $T$ and the behavior described by (3), (4) is therefore to be considered emergent.

In conclusion, the purpose of this note is to highlight a particular relation between masses and lifetimes of elementary particles. While this relation seems perfectly consistent with what is known about particles and conforms to the general framework of the Standard Model, to our knowledge its emergence from this framework has never been previously discussed. Nevertheless, we believe that

only through the analysis of this emergence, to be left to further studies, the actual relevance of this result can be clarified.

**Funding:** This research received no external funding.

**Acknowledgments:** The author thanks the reviewers for their suggestions which have permitted an important improvement of this paper.

**Conflicts of Interest:** The author declares no conflict of interest.

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
