# Peer review of "Power Laws and Elementary Particle Decays"

_sci, doi:10.3390/sci2010017_

Round 1

Reviewer 1 Report

The paper has interesting results and its publication is recommended. However, I consider some changes necessary to improve the reader's understanding.
In particular:
a) 
figure 1 is useless being a duplicate of table1 and can lead to misunderstanding.
b) 
it may be useful to argue in more detail the interpretation of as an information measure (eq. 5).
c) 
I reworked the analysis of chap.3, calculating n from eq.5 and θ from eq.2. From my analysis it emerges that only in the case of θ = k1 Mk2 the data in table 1 form three well separated clusters. I consider it necessary that the author describe his analysis in detail.

Author Response

The paper has interesting results and its publication is recommended. However, I consider some changes necessary to improve the reader's understanding. In particular: a) figure 1 is useless being a duplicate of table1 and can lead to misunderstanding. b) it may be useful to argue in more detail the interpretation of n as an information measure (eq. 5). c) I reworked the analysis of chap.3, calculating n from eq.5 and θ from eq.2. From my analysis it emerges that only in the case of θ = k1 Mk2 the data in table 1 form three well separated clusters. I consider it necessary that the author describe his analysis in detail. I thank the reviewer for his relevant comments that have helped me in the preparation of this second version. I kept Figure 1, but I added Figure 3 according to the reviewer indications. From the contrast between Figures 2 and 3 the reader can get an idea of the difference between the two correlations. Furthermore, Figure 3 offers a synopsis of the three families. I added a clear demonstration that index n is an information measure, and equation (5) is the representation of its left hand member in terms of the average information relative to n.

Reviewer 2 Report

The reported work "Power laws and elementary particle decays" by Leonardo Chiatti concerns correlation between the lifetime and the rest energy of the unstable particle states with a lifetime greater than the zeptosecond. The author claim that the set of a such states seems to be divided into three different "families", in each of which the two quantities can be correlated through a remarkably accurate power law. The mechanism responsible for this, however, remains to be explained. Since the work contains some elements of novelty, I recommend publish it unchanged.

Author Response

The reported work "Power laws and elementary particle decays" by Leonardo Chiatti concerns correlation between the lifetime and the rest energy of the unstable particle states with a lifetime greater than the zeptosecond. The author claim that the set of a such states seems to be divided into three different "families", in each of which the two quantities can be correlated through a remarkably accurate power law. The mechanism responsible for this, however, remains to be explained. Since the work contains some elements of novelty, I recommend publish it unchanged. Dear colleague Thanks to have reviewed this paper and for the encouragement.

Reviewer 3 Report

This paper presents an interesting set of patterns in the decay rates of elementary  particles which prevail over many orders of magnitude in their half-lives. Given the large number of decays for which they quote data, this pattern is certainly not accidental, and does not have an obvious explanation that I can see.

My general feeling in such cases is that the data deserve to be published in the hope that eventually their availability would stimulate others to pursue an explanation inmate detail.

My recommendation is that this paper be accepted for publication, subject to minor style or language changes that the editors may wish to suggest.

Reviewer 4 Report

Particle lifetimes are well theoretically predicted in the framework of the Standard Model and found to depend on particle masses. So any systematic correlation between these two quantities, if it exists, can and should be directly derived and interpreted in term of first principles.

The work presented here first introduces arbitrarily defined variables and correlation among them, then arbitrarily includes or excludes particles to be considered in the study. In particular there is not any really physically based motivation to exclude the neutron and all particles with lifetime shorter than 1 zeptosecond.

The fitting procedure bringing to the final results on the power laws parametrization is not properly reported and quantified in its soundness.  

The overall impression is that this study is characterize by a strong “fine tuning” of inputs and ansatz, so it is really hard to agree on the final conclusions.

Author Response

Particle lifetimes are well theoretically predicted in the framework of the Standard Model and found to depend on particle masses. So any systematic correlation between these two quantities, if it exists, can and should be directly derived and interpreted in term of first principles. The work presented here first introduces arbitrarily defined variables and correlation among them, then arbitrarily includes or excludes particles to be considered in the study. In particular there is not any really physically based motivation to exclude the neutron and all particles with lifetime shorter than 1 zeptosecond. The fitting procedure bringing to the final results on the power laws parametrization is not properly reported and quantified in its soundness. The overall impression is that this study is characterize by a strong “fine tuning” of inputs and ansatz, so it is really hard to agree on the final conclusions. I cannot agree with the reviewer's comments. First of all, they completely miss the point. There is no doubt that "particle lifetimes are well theoretically predicted in the framework of the Standard Model". This is repeated several times in the text of the communication. But a cursory look at the handbooks shows that the masses of the products enter the calculation, so the lifetime of a particle depends on all its modes of decay and on the masses of decay products in each channel. So it is not immediately obvious how a relation such as (3), (4) between the lifetime of a particle and the single mass of that same particle can arise. These relationships do not describe, as do the usual applications of the Standard Model, the details of the single decay channel of a single particle. They show that the net of all channels relative to all the particles exhibits a behavior describable by scale laws. In other words, the three "families" are sub-nets with a self-similar behavior as the mass varies. To understand the real state of affairs, an analogy can be useful. Suppose we are interested in the study of the hourly trend of traffic in a large city. For the purposes of such a study it is almost completely unnecessary to know the precise mechanical mode of operation of the individual engine of each individual car. And this despite the fact that it is precisely this operation that makes the phenomenon under investigation possible. In the analogy, the mechanics of the single engine is the Standard Model and the traffic is the correlations (3), (4). The phenomenon of traffic cannot be formulated in the variables of mechanics of the single engine; for this reason new variables are required (entry times in schools, location of shops and production centers, etc.). These "new" variables are those defined by (1), (2). No arbitrary particle selection is made in the study. The neutron does not adapt to the scheme. But it should be kept in mind that from the point of view of its half-life the neutron is a "monster" among the elementary particles. The ratio between this half-life and that of the muon (the particle with slowest decay after the neutron) is 1 to a billion. Of course this does not mean that the neutron "engine" is different. The interaction involved in both decays is in fact always the weak one. Particles with a lifetime less than 1 zs are (essentially) the radially or rotationally excited states of the hadronic states considered in the study; their decay is mediated by strong interaction and lifetimes range from 10-21 to 10-24 s. They can therefore be divided into three bins of one order of magnitude each. Since the distribution of the states is roughly homogeneous in each of the three bins and the mass and the composition of flavor are distributed in a roughly homogeneous way in each bin, it is evident the absence of any dependence of T from M with the precision of one or two bins. All this is briefly explained in the text. A plot in the style of Figure 1 extended to these states can be found in arXiv:hep-ph/0506033, Figure 2 and Table (personally, however, I do not agree with the author's arguments). As regards the fitting procedure, by plotting n (fourth column of Table 1) as a function of M it is possible to immediately obtain the three curves relating to the three families. The three families are also visible in Figure 2 of the ref. [2]. There the neutron is included.

Reviewer 5 Report

see below

Round 2

Reviewer 1 Report

Considering the changes made by the author satisfactory I recommend the publication of the paper.

Author Response

Considering the changes made by the author satisfactory I recommend the publication of the paper. Dear colleague Thanks to have reviewed this paper and for data reworking.

Reviewer 4 Report

No additional comment wrt previous review.

Author Response

No additional comment wrt previous review. I am sincerely sorry that the comments reported in the previous review did not convince the reviewer. In any case, I thank him for taking the time to evaluate this article.